# Structure of a step II catalytically activated spliceosome from *Chlamydomonas reinhardtii*

Yichen Lu[1,2,3,4,5], Ke Liang[2,3,4,5] & Xiechao Zhan [2,3,4] ✉

## Abstract

**Pre-mRNA splicing, a fundamental step in eukaryotic gene expression, is executed by the spliceosomes. While there is extensive knowledge of the composition and structure of spliceosomes in yeasts and humans, the structural diversity of spliceosomes in non-canonical organisms remains unclear. Here, we present a cryo-EM structure of a step II catalytically activated spliceosome (C\* complex) derived from the unicellular green alga *Chlamydomonas reinhardtii* at 2.6 Å resolution. This *Chlamydomonas* C\* complex comprises 29 proteins and four RNA elements, creating a dynamic assembly that shares a similar overall architecture with yeast and human counterparts but also has unique features of its own. Distinctive structural characteristics include variations in protein compositions as well as some noteworthy RNA features. The splicing factor Prp17, with four fragments and a WD40 domain, is engaged in intricate interactions with multiple protein and RNA components. The structural elucidation of *Chlamydomonas* C\* complex provides insights into the molecular mechanism of RNA splicing in plants and understanding splicing evolution in eukaryotes.**

**Keywords** *Chlamydomonas* Spliceosome; C\* Complex; U6 snRNA; Cryo-EM
**Subject Categories** Plant Biology; RNA Biology; Structural Biology

## Introduction

Pre-mRNA splicing, a fundamental and evolutionarily conserved aspect of eukaryotic gene expression, threads seamlessly through the rich tapestry of life, from unicellular organisms to intricate multicellular entities, transcending the boundaries between the plant and animal kingdoms (Kastner et al, 2019; Lorkovic et al, 2000; Plaschka et al, 2019; Shi, 2017a; Wan et al, 2020). Orchestrated by spliceosomes, RNA splicing choreographs the removal of non-coding introns and the precise joining of coding exons through a coordinated ballet of branching and exon ligation reactions (Shi, 2017b). The spliceosome is among the most dynamic molecular machines in eukaryotic cells, relying on the complex interplay between myriad RNA molecules and ribonucleoproteins (RNP) as they pass through a sequence of highly regulated conformational states (Yan et al, 2019).

During splicing, three conserved intronic elements—the 5′-splice site (5′SS), the branch site (BS), and the 3′-splice site (3′SS)—play essential roles in the sequential reactions (Wan et al, 2020). For the branching reaction, the BS attacks the 5′SS, resulting in a free 5′-exon and an intron lariat-3′-exon intermediate, with the formation of the catalytic step-I spliceosome (C complex) (Bertram et al, 2020; Wilkinson et al, 2021; Zhan et al, 2018). The C complex is remodeled into the step II catalytically activated spliceosome (C\* complex), representing a functional state poised and awaiting the exon ligation reaction (Zhan et al, 2022). In the C\* complex, the 3′-end of 5′-exon attacks the 5′-end of 3′-exon, generating a ligated exon and an intron lariat (Fica et al, 2019; Zhang et al, 2019).

Detailed structural characterization of spliceosomes from both yeasts and humans has provided profound insights into the mechanism of RNA splicing (Kastner et al, 2019; Plaschka et al, 2019; Yan et al, 2019), yet many questions remain regarding the structural diversity of spliceosomes across other different species, like in plants. Do the composition and the catalytic center have any different features in plants? Are there any specific splicing factors and regulation mechanisms in plants? The unicellular green alga *Chlamydomonas reinhardtii* is a valuable non-canonical model organism for the study of RNA splicing, and it has garnered attention for its role in studies related to photosynthesis and flagellar biology (Foster and Smyth, 1980; Polin et al, 2009; Rochaix, 1995). The appeal of *Chlamydomonas* lies not only in its evolutionary distance from canonical model organisms but also in its ability to thrive under diverse environmental conditions (Collins and de Meaux, 2009). Its phylogenetic position and unique genomic features make it an intriguing subject for investigating the evolution and structural divergence of splicing machinery.

In this study, we purified the endogenous *Chlamydomonas* spliceosome and determined the structure of the *Chlamydomonas* C\* complex using cryo-electron microscope (cryo-EM) analysis at a resolution of 2.6 Å. This revealed structural similarities between the *Chlamydomonas* C\* complex and its counterparts in yeasts and humans. Our work deepens the general understanding of splicing machinery in non-canonical organisms and also highlights the conserved nature of critical splicing components across evolutionary landscapes.

[1]College of Life Sciences, Fudan University, Shanghai 200433, China. [2]Westlake Laboratory of Life Sciences and Biomedicine, 18 Shilongshan Road, Hangzhou, Zhejiang 310024, China. [3]Key Laboratory of Structural Biology of Zhejiang Province, School of Life Sciences, Westlake University, 18 Shilongshan Road, Hangzhou, Zhejiang 310024, China. [4]Institute of Biology, Westlake Institute for Advanced Study, 18 Shilongshan Road, Hangzhou, Zhejiang 310024, China. [5]These authors contributed equally: Yichen Lu, Ke Liang. ✉E-mail: zhanxiechao@westlake.edu.cn

# Results

## Spliceosome isolation and cryo-EM analysis

To purify the *Chlamydomonas* spliceosomes, we generated two knock-in (KI) cell strains with a 3x Flag tag added to the C-terminus of an authentic Cdc5L (Cdc5L-Flag strain) or Prp19 (Prp19-Flag strain) gene locus using CRISPR/Cas9 mediated gene editing. Following validation by PCR and DNA sequencing, the two KI strains showed normal growth, comparable to wild-type (WT) strains, suggesting that the introduced tags had not adversely affected the function of the spliceosomes. The endogenous *Chlamydomonas* spliceosomes were independently isolated by affinity purification and glycerol gradient centrifugation with chemical crosslinking from the Cdc5L-Flag and Prp19-Flag strains (Fig. EV1A–C). The resulting purified sample was subjected to cryo-EM analysis (Fig. EV1D–G), yielding two high-resolution reconstructions, both identified as the C* complex at 2.6 Å without obvious differences (Figs. EV2–4).

The final cryo-EM maps of the *Chlamydomonas* C* complex display local resolutions of 2.5–3.0 Å in the core region, enabling the identification of protein components and the assignment of RNA elements (Fig. EV5; Appendix Figs. S1–3 and Tables S1–3). The resolutions of the local EM maps at the periphery are comparatively moderate due to the dynamic nature of the complex and are further refined to improve modeling.

## Overall structure of the *Chlamydomonas* C* complex

The structure of the *Chlamydomonas* C* complex reveals an intricate, extended, and asymmetric assembly. This complex comprises a total of 29 proteins, each playing a specific role in the splicing process. Key constituents are distributed across six functional sub-units, delineating the complex into discrete components with specific roles (Appendix Table S2). In the *Chlamydomonas* C* complex, the first sub-unit U5 small nuclear RNP (U5 snRNP) serves as a foundational building block, with ten integral proteins including Prp8, Snu114, U5-40K and seven U5 Sm-ring proteins. The sub-unit Nineteen Complex (NTC) features eight proteins: Spf27, Cdc5L, Cwc15, PLRG1, and four copies of Prp19. The NTC-Related (NTR) complex contains six proteins: Syf2, Syf3, Bud31, RBM22, SKIP, and PPIL1. Three splicing factors—Prp17, Cwc21, and Cwc22 and two intron binding proteins (IBP)—Syf1 and Aquarius participate in the splicing process as other two sub-units. Finally, four crucial RNA elements—U2 snRNA, U5 snRNA, U6 snRNA, and pre-mRNA complete the essential constituents of the molecular machinery (Fig. 1A). The asymmetric assembly and distribution of these components highlight the complexity and functional specialization of the *Chlamydomonas* spliceosome, providing valuable insights into the intricacies of the splicing process in this organism.

## Structural comparison with *S. cerevisiae* and human C* complexes

Structural study of the *Chlamydomonas* C* complex allows a comparative analysis with the *S. cerevisiae* and human C* complexes (Yan et al, 2017; Zhang et al, 2017), unveiling both similarities and distinctions (Fig. 1B; Appendix Table S3). Compared to *S. cerevisiae* (Yan et al, 2017), the splicing factors Prp18 and Prp22 are not observed in the *Chlamydomonas* C* complex; the sequences of two NTR components Ecm2 and Cwc2 in yeast correspond to *Chlamydomonas* RBM22 (Rasche et al, 2012). However, the *Chlamydomonas* C* complex contains the additional components Aquarius and U5-40K, which lack the genes in *S. cerevisiae*. Besides, the precise modeling of the U2 snRNP core, the N-terminal domain of Cwc22, and the C-terminal WD40 domain of Prp19 remains challenging due to their flexibility, which emphasizes the dynamic nature of the *Chlamydomonas* C* complex.

Despite these variances, the overall architecture of the *Chlamydomonas* C* complex bears a striking resemblance to that of the human C* complex (Zhang et al, 2017), extending to the core components and their spatial arrangement. However, divergences arise in the human C* complex, which contains additional splicing factors, including PPIE, PRKRIP1, BRR2, PRP22, SLU7, FAM32A, and the exon junction complex (EJC) (Fig. 1C). Comparative analysis highlights both evolutionary conservation and species-specific variations in the composition of the C* complex, demonstrating the versatility of the splicing machinery across eukaryotes.

## Structural features of the RNA elements

Four RNA elements are assigned in the *Chlamydomonas* C* complex: U2, U5 and U6 snRNAs, and pre-mRNA (Fig. 2A). The released 5′-exon is anchored on loop I of the U5 snRNA (Fig. EV5A), which is highly conserved among the yeast, *Chlamydomonas*, and human (Appendix Fig. S4). The 5′SS is linked to the BS, and its ensuing sequences form an extended U6/5′ SS duplex with the U6 snRNA. The U2 snRNA forms Helix II/I with the U6 snRNA and the U2/BS duplex with the BS (Fig. EV5B). In addition, two catalytic and four structural metal ions are identified at the active site, which is attributed to the internal-stem-loop (ISL) of the U6 snRNA (Fig. EV5C). Notably, a γ-monomethyl phosphate cap is structurally ascertained at the 5′-end of the U6 snRNA (Figs. 2A and EV5D). Altogether, the complex RNA assembly is stabilized by multiple intra/inter-molecular base-pairings and surrounding proteins (Fig. 2B).

The resolved structure of RNA in *Chlamydomonas* C* resembles that in *S. cerevisiae* C* except for the U6/5′SS duplex and the 5′-stem loop (5′-SL) of the U6 snRNA (Yan et al, 2017) (Fig. 3A). The U6/5′SS duplex extends to 16 nucleotides in *Chlamydomonas* C* compared to the eight nucleotides in *S. cerevisiae* (Yan et al, 2017) (Fig. 3B). This structural feature is also found in human C* and other post-activated spliceosomal states (Zhan et al, 2018; Zhang et al, 2017; Zhang et al, 2018; Zhang et al, 2019). The ACAGA box is highly conserved among the yeast, *Chlamydomonas*, and human (Appendix Fig. S5). The organization of RNA in *Chlamydomonas* C* is almost identical to that in human C*, except for the 5′-SL of U6 snRNA (Fig. 3C). Compared to *S. cerevisiae*, the 5′-SL of U6 snRNA in *Chlamydomonas* is much shorter. The first γ-monomethyl phosphate capped nucleotide G1 is base-paired with the nucleotide C12 in *Chlamydomonas*, while the G1 in *S. cerevisiae* is base-paired with the nucleotide C25 (Fig. 3D). In human C*, the 5′-SL of U6 snRNA is formed between the paired nucleotides G1 and C19 (Fig. 3E). The structural variance of 5′-SL among yeast, *Chlamydomonas*, and human also conforms to the sequence diversity at the 5′-end of U6 snRNA (Appendix Fig. S5).

Although the RNA elements at the active site adopt a similar conformation to those in *S. cerevisiae* and human C* (Yan et al,

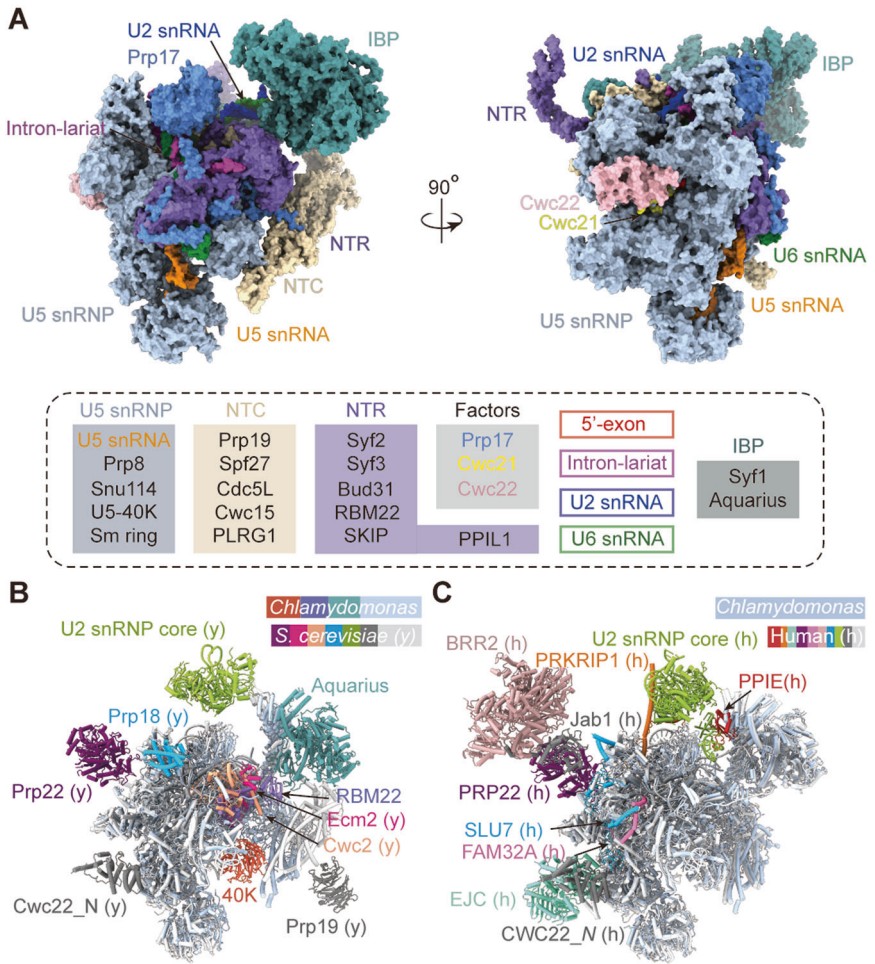

**Figure 1.  Cryo-EM structure of the *Chlamydomonas* C˙ complex.**

(A) Overall structure of the *Chlamydomonas* C˙ complex. Two perpendicular views are shown. All components are presented in colored surface view and tabulated below the images. The *Chlamydomonas* C˙ complex contains 29 proteins, with ten in U5 snRNP (Prp8, Snu114, U5-40K, and U5 Sm-ring proteins), eight in NTC (four copies of Prp19, Spf27, Cdc5L, Cwc15, and PLRG1), six in NTR (Syf2, Syf3, Bud31, RBM22, SKIP, and PPIL1), three splicing factors (Prp17, Cwc21, and Cwc22), two intron binding proteins (Syf1 and Aquarius) and four RNA elements (U2 snRNA, U5 snRNA, U6 snRNA, and pre-mRNA). All structural images in this manuscript were prepared using ChimeraX (Pettersen et al, 2021). (B) Structural comparison between the *Chlamydomonas* and *S. cerevisiae* C˙ complexes. Compared to *S. cerevisiae*, the *Chlamydomonas* C˙ complex lacks Prp18, Prp22, Ecm2, and Cwc2, but contains Aquarius, RBM22, and U5-40K. The U2 snRNP core, the N-terminal domain of Cwc22, and the C-terminal WD40 domain of Prp19 cannot be modeled due to their flexibility. (C) Structural comparison between the *Chlamydomonas* and human C˙ complexes. The overall architecture of the *Chlamydomonas* C˙ complex is similar to that of human C˙ complex, except that human C˙ contains extra splicing factors, including PPIE, PRKRIP1, BRR2, PRP22, SLU7, FAM32A, and EJC.

2017; Zhang et al, 2017), there are some distinctive features that reveal the structural specificity of *Chlamydomonas*. The key nucleotide U67, of the U6 snRNA, is base-paired with the nucleotide G20 of the U2 snRNA, and two catalytic metal ions are coordinated by the phosphate group of U67, with M1 sharing a distance of 3.2 Å to the 3′-OH of the last nucleotide G-1 of the 5′-exon (Fig. 3F). The nucleotide G21 of the U2 snRNA, which corresponds to the G20 base-paired to C55 of U6 snRNA in human C˙ (Fig. 3G), is base-paired with C48 of U6 snRNA. Compared to *Chlamydomonas*, the key nucleotide U74 of U6 snRNA is bulged out and not base-paired in human C˙. Notably, the sequences of 3′ SS and 3′-exon in *Chlamydomonas* C˙ are too flexible to be assigned, likely due to their heterogeneity from the endogenous purification.

## Structural features of the splicing factor Prp17

The splicing factor Prp17 comprises four fragments (fragments I–IV) and a WD40 domain. It is unambiguously assigned in the *Chlamydomonas* C˙ complex and adopts an extended overall conformation with each fragment or domain contacting multiple protein and RNA components (Fig. 4A,B). For fragment I (residues: 47–93), the N-terminal loop interacts with the NTC core (Prp19 subcomplex, Spf27, and the C-terminus of Cdc5L) and the middle portion is sandwiched between the U5-40K/C-terminal Smb and PPIL1 (Fig. 4C; Appendix Fig. S2A). Notably, the residue Pro89 is inserted into the catalytic pocket of PPIL1, which probably represents the potential target of the peptidyl-proline isomerase activity of PPIL1 (Zhan et al, 2018) (Appendix Fig. S2A). The C-terminal portion of

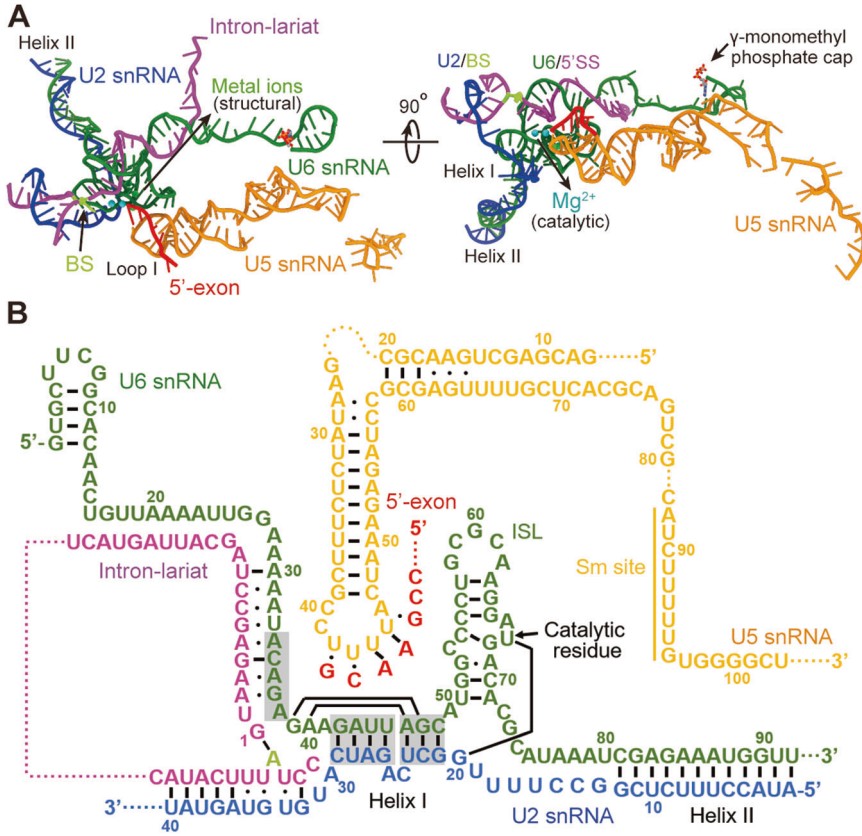

**Figure 2. Structure of the RNA elements.**

(A) Overall structure of the RNA elements in the *Chlamydomonas* C˙ complex. Two perpendicular views are shown, and the color-coding scheme is preserved throughout this manuscript. Four RNA elements are assigned in the *Chlamydomonas* C˙ complex: U2, U5, U6 snRNAs, and pre-mRNA. The released 5'-exon is anchored on loop I of the U5 snRNA. The 5'SS is linked to BS, and its ensuing sequences form an extended duplex with the U6 snRNA. The U2 snRNA forms Helix II/I with U6 snRNA and the U2/BS duplex with BS. Two catalytic and four structural metal ions are identified at the active site. A γ-monomethyl phosphate cap is identified at the 5'-end of U6 snRNA. (B) Summary of the base-pairing interactions among the RNA elements. Canonical Watson-Crick and non-canonical base-pairing interactions are identified by solid lines and dots, respectively.

fragment I loosely contacts RBM22. Moreover, the fragment I segment is absent in the *S. cerevisiae* Prp17 gene, and it is partially resolved in the human C˙ complex, showing interactions with Smb, U5-40K, and PPIL1 (Zhan et al, 2022). These interactions are similar to those observed in *Chlamydomonas*, revealing the conserved role of Prp17 in spliceosome assembly and function across eukaryotes. For fragment II (residues: 101–146), a β-strand at the N-terminal portion is parallel with another β-strand of SKIP, and the C-terminal portion interacts with Prp8, U5-40K, RBM22, Bud31, and U6 snRNA (Fig. 4D; Appendix Fig. S2B).

Fragment III (residues: 165–203) of Prp17 contacts the 5'-SL of the U6 snRNA and stabilizes its conformation. The C-terminal loop of fragment III interacts with Bud31 (Fig. 5A; Appendix Fig. S2C). In addition, the first nucleotide G1 of U6 snRNA is γ-monomethyl phosphate capped, which contributes to bridging the interaction between the 5'-SL and fragment III. This interaction features multiple hydrogen bonds: Lys166 and the base group of U2; Arg171 and the phosphate group of G3; Lys172 and the phosphate group of G8; Arg178 and the phosphate group of G3; and Arg181 and the γ-phosphate group of G1 (Fig. 5B). Fragment IV (residues: 211–230) represents an α-helix and covers the U6/5'SS duplex, which is likely

to assist in stabilizing the extended duplex (Fig. 5C). At the divergence of the U6/5'SS duplex, the nucleotide A21 of the U6 snRNA is recognized by the zinc-finger motif of RBM22 and stacks against the residue Phe168. The nucleotides A20 and A21 of the U6 snRNA stack against the aromatic residues Tyr182 and Phe168 of RBM22, respectively (Fig. 5D). The WD40 domain of Prp17 is positioned above the U6/5'SS duplex and loosely contacts the U2/BS duplex, Cdc5L, and the RNase-H-like (RH) domain of Prp8 (Fig. 5E; Appendix Fig. S2D).

The extended Prp17 in *Chlamydomonas* C˙ is better resolved compared to its counterparts in yeast and human C˙ complexes (Yan et al, 2017; Zhan et al, 2022). This improved resolution is attributed to its function and the sequence variability of the 5'-end of U6 snRNA in Chlamydomonas (Appendix Fig. S5). Specifically, the shorter U6 snRNA in *Chlamydomonas* allows for the proper accommodation of the N-terminus of Prp17. This structural feature is consistent with previous studies suggesting that Prp17 plays a role as a step II splicing factor (Jones et al, 1995; Zhou and Reed, 1998). While these structural insights provide intriguing possibilities, further biochemical and genetic analyses are needed to directly link these features of Prp17 to its adaptive function in *Chlamydomonas*.

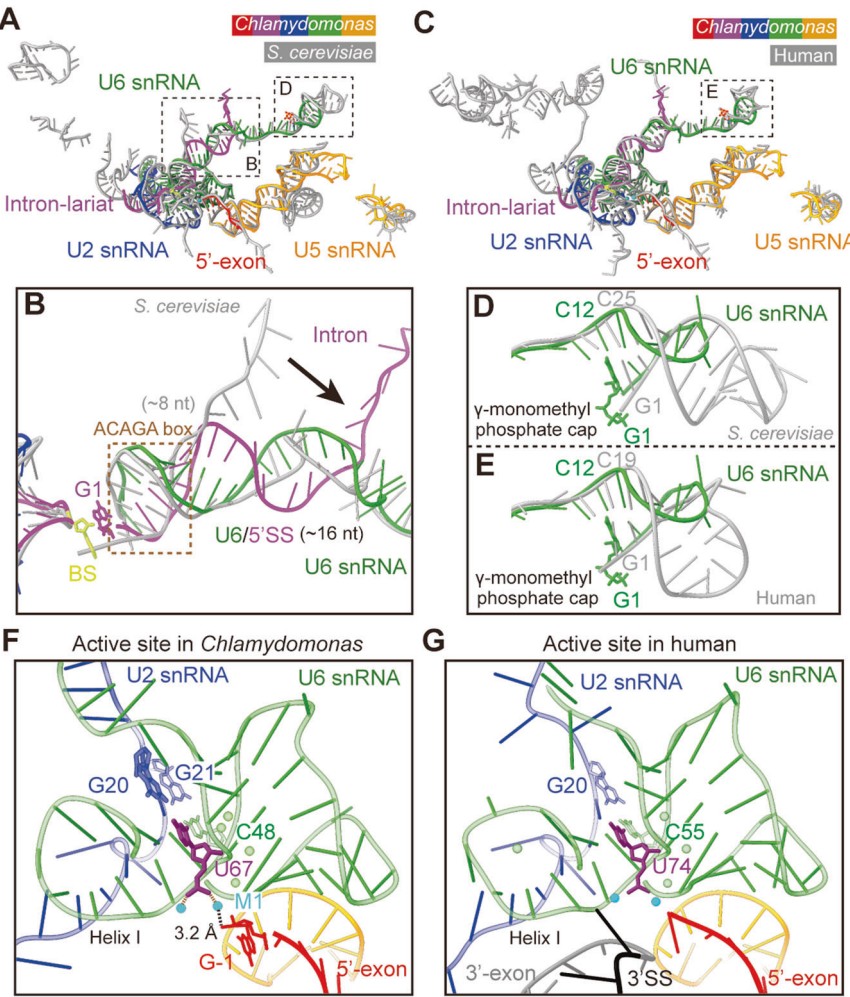

**Figure 3. Structural comparison of the RNA elements.**

(A) Structural comparison of the RNA elements between the *Chlamydomonas* and *S. cerevisiae* C* complexes. The resolved structure of RNA elements in *Chlamydomonas* C* is similar to that in *S. cerevisiae* C*, except for the U6/5′SS duplex and the 5′-SL of U6 snRNA. (B) A close-up view of the U6/5′SS duplex. The U6/5′SS duplex extends to 16 nucleotides in *Chlamydomonas* C* compared to eight nucleotides in *S. cerevisiae*. (C) Structural comparison of the RNA elements between the *Chlamydomonas* and human C* complexes. The modeled structure of RNAs in *Chlamydomonas* C* is almost identical to that in human C*, except for the 5′-SL of U6 snRNA. (D) A close-up view of the 5′-SL of U6 snRNA in *S. cerevisiae* and *Chlamydomonas*. The 5′-SL of U6 snRNA in *Chlamydomonas* is much shorter than that in *S. cerevisiae*. The first γ-monomethyl phosphate capped nucleotide (G1) is base-paired with the nucleotide C12 in *Chlamydomonas*, while, in *S. cerevisiae*, G1 is base-paired with the nucleotide C25. (E) A close-up view of the 5′-SL of U6 snRNA in human and *Chlamydomonas*. In human, the 5′-SL of U6 snRNA is formed between the paired nucleotides G1 and C19. (F) A close-up view of the active site in *Chlamydomonas* C*. The key nucleotide U67 of U6 snRNA is base-paired with the nucleotide G20 of U2 snRNA, and two catalytic metal ions are coordinated by the phosphate group of U67, with M1 sharing a distance of 3.2 Å to the 3′-OH of the last nucleotide G-1 of 5′-exon. The nucleotide G21 of U2 snRNA, which corresponds to the G20 base-paired to C55 of U6 snRNA in human C*, is base-paired with C48. (G) A close-up view of the active site in human C*. The key nucleotide U74 of U6 snRNA bulges out and is not base-paired. The sequences of 3′SS and 3′-exon are docked into the active site in human C*.

# Discussion

Structural analysis of the *Chlamydomonas* C* complex sheds light on critical aspects of pre-mRNA splicing machinery, expanding our understanding of its conservation and diversification during eukaryotic evolution. Here, we delve into our key findings, draw comparisons with *S. cerevisiae* and human spliceosomes, and highlight the distinctive features observed in the *Chlamydomonas* spliceosome.

The *Chlamydomonas* C* complex is a dynamic assembly of 29 proteins and four essential RNA elements that underscores the evolutionary conservation of the splicing machinery. While sharing an overall architectural similarity with *S. cerevisiae* and human C* complexes, our study identifies a number of specific structural differences. The absence of Prp18, Prp22, Ecm2, and Cwc2, and the presence of additional components such as Aquarius, RBM22, and U5-40K in *Chlamydomonas* highlight both its common ancestry with *S. cerevisiae* and human C* and species-specific variations. Examination of the RNA elements in the *Chlamydomonas* C* complex reveals several distinctive features. The extended U6/5′SS duplex, similar to that in the human spliceosome (Zhan et al, 2022), contrasts with its shorter counterpart in *S. cerevisiae*. The ACAGA

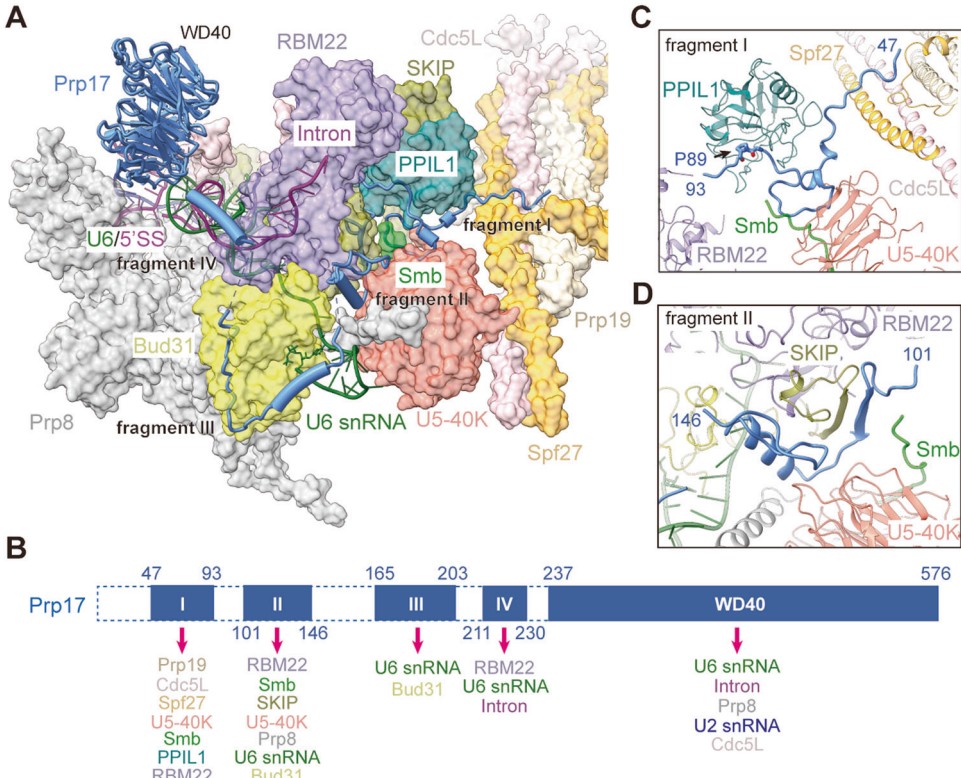

Figure 4. Structural features of the splicing factor Prp17.

(A) The structure of the splicing factor Prp17 (colored blue) in the *Chlamydomonas* C* complex. The four fragments (fragments I–IV) and the WD40 domain of Prp17 are unambiguously assigned, contacting multiple protein and RNA components. (B) Schematic diagram of the domain feature of modeled Prp17 with its interacting elements listed below. (C) A close-up view of fragment I (residues: 47–93). The N-terminal loop of fragment I interacts with the NTC core (Prp19 subcomplex, Spf27, and the C-terminus of Cdc5L). The middle portion of fragment I is sandwiched between U5-40K/the C-terminal Smb and PPIL1. The C-terminal portion of fragment I loosely contacts RBM22. (D) A close-up view of fragment II (residues: 101–146). At the N-terminal portion, a β-strand parallels another β-strand of SKIP. The C-terminal portion interacts with Prp8, U5-40K, RBM22, Bud31, and U6 snRNA.

box, a highly conserved motif (Li and Brow, 1996), aligns across yeast, *Chlamydomonas*, and human. Yet, variations in the 5′-SL of U6 snRNA introduce structural diversity. The nuanced interactions of U67, G20, and G21 of U6 snRNA with U2 snRNA at the active site of the *Chlamydomonas* C* complex reveal the specificity of the splicing process in this organism. The splicing factor Prp17 exhibits a complex network of interactions within the *Chlamydomonas* C* complex. Fragments I–IV and the WD40 domain of Prp17 play pivotal roles in stabilizing interactions with other proteins and RNA elements. Noteworthy is the insertion of Pro89 into the catalytic pocket of PPIL1, indicating the potential targets for peptidyl-proline isomerase activity.

The diversity observed in *Chlamydomonas* spliceosomes suggests functional adaptations to unique environmental challenges. The flexible sequences of 3′SS and the 3′-exon may reflect substrate heterogeneity arising from endogenous purification. These structural features underscore the intricate dance of the spliceosome and prompt further exploration into how *Chlamydomonas* responds to environmental cues. Our study emphatically positions *Chlamydomonas* as a key player in the evolutionary and structural aspects of splicing machinery. The conservation of critical splicing

components across evolutionary landscapes signifies the broader implications of our work.

The insights gained from the *Chlamydomonas* spliceosome structure offer valuable perspectives for understanding splicing machinery in diverse plant species. While our study focuses on *Chlamydomonas*, it provides a foundation for comparative studies across plants. Plants exhibit remarkable diversity in splicing regulation and alternative splicing patterns (Syed et al, 2012), which contribute to their adaptation to diverse environmental conditions. By elucidating structural variations and unique features in the *Chlamydomonas* spliceosome, we lay the groundwork for comparative analyses with other plant species, potentially uncovering commonalities and distinctions in splicing machinery.

In conclusion, the structural characterization of the *Chlamydomonas* C* complex adds a vibrant thread to the tapestry of splicing machinery as it has evolved and also prompts new questions and avenues for exploration. The intricate interplay of proteins and RNA elements in *Chlamydomonas* contributes to our understanding of the splicing process in this organism, providing a foundation for future studies on splicing mechanisms in other plants.

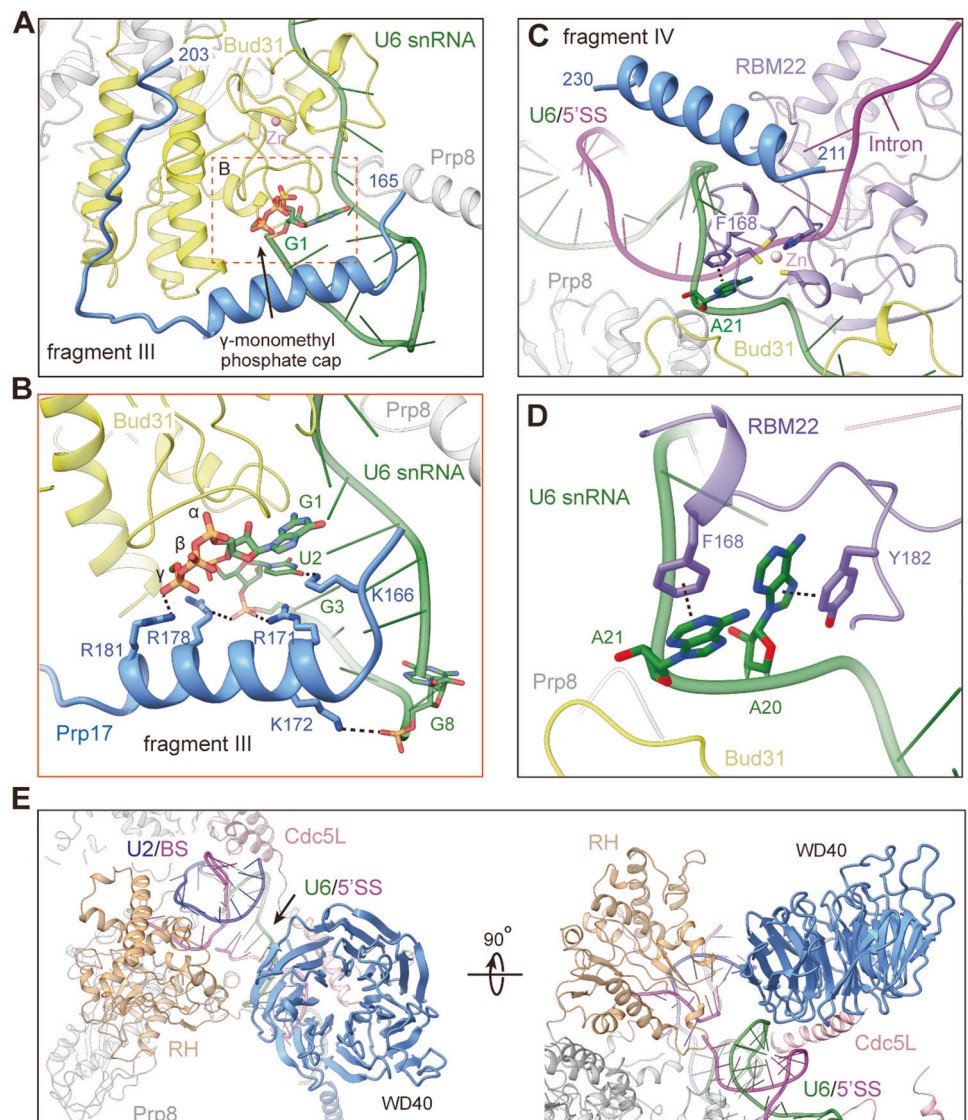

**Figure 5. Structural features around the Prp17 C-terminus.**

(A) Structural features of fragment III (residues: 165–203) of Prp17. The N-terminal helix contacts the 5′-SL of U6 snRNA and stabilizes the conformation of the 5′-SL. The C-terminal loop interacts with Bud31. (B) A close-up view of the interface between 5′-SL and fragment III. The interaction between 5′-SL and fragment III features multiple hydrogen bonds: Lys166 and the base group of U2; Arg171 and the phosphate group of G3; Lys172 and the phosphate group of G8; Arg178 and the phosphate group of G3; and Arg181 and the γ-phosphate group of G1. (C) Structural features around fragment IV (residues: 211–230) of Prp17. This helix covers the U6/5′SS duplex, which probably helps to stabilize the extended duplex. At the divergence of the U6/5′SS duplex, the nucleotide A21 of U6 snRNA is recognized by the zinc-finger motif of RBM22 and stacks against the residue Phe168. (D) A close-up view around the nucleotide A21 of U6 snRNA. The nucleotides A20 and A21 of U6 snRNA stack against the aromatic residues Tyr182 and Phe168 of RBM22, respectively. (E) Structural features around the WD40 domain of Prp17. Two perpendicular views are shown.

# Methods

## Reagents and tools table

| Reagent/Resource | Reference or Source | Identifier or Catalog Number |
|---|---|---|
| **Experimental Models** | | |
| *Escherichia coli* strain DH5α | Thermo Fisher Scientific | Cat# 18265017 |
| *Escherichia coli* strain BL21 (DE3) | Thermo Fisher Scientific | Cat# EC0114 |

| Reagent/Resource | Reference or Source | Identifier or Catalog Number |
|---|---|---|
| *Chlamydomonas reinhardtii* CC-4533 strain | (Zhang et al, 2014) | CC-4533 |
| *Chlamydomonas reinhardtii* Cdc5L-Flag KI strain | This study | N/A |
| *Chlamydomonas reinhardtii* Prp19-Flag KI strain | This study | N/A |
| **Recombinant DNA** | | |
| Plasmid for His-SV40NLS-SpCas9-npNLS | This study | N/A |

| Reagent/Resource | Reference or Source | Identifier or Catalog Number |
|---|---|---|
| **Oligonucleotides and other sequence-based reagents** | | |
| Guide sequence (5′-3′) for Cdc5L-Flag KI: ACTAGCGCAGAAGCGGGCGG | This study | N/A |
| Guide sequence (5′-3′) for Prp19-Flag KI: CCACATAACCGCTTAAGCGG | This study | N/A |
| **Chemicals, Enzymes and other reagents** | | |
| Glutaraldehyde (EM grade) | Sigma-Aldrich | CAT#G5882 |
| Uranyl acetate | Sigma-Aldrich | CAT#73943-25G |
| SYBR®Gold | Thermo Fisher Scientific | CAT#S11494 |
| Glycogen | Fermentas | CAT#R0551 |
| Proteinase K | Amresco | CAT#Amresco0706 |
| Aprotinin | Amresco | CAT#E429 |
| Pepstatin | Amresco | CAT#J583 |
| Leupeptin | Amresco | CAT#J580 |
| Phenylmethyl sulfonyl fluoride | Solarbio | Cat# P8340 |
| Anti-Flag M2 affinity gel | Sigma | CAT#A2220 |
| LB medium | BD | CAT#214966 |
| **Software** | | |
| EPU 2.12.0 | Thermo Fisher Scientific | N/A |
| MotionCor2 | (Zheng et al, 2017) | https://emcore.ucsf.edu/ ucsf-software; RRID:SCR_016499 |
| Gctf | (Zhang, 2016) | https://github.com/JackZhang-Lab/GCTF |
| cryoSPARC v3.3.2 | (Punjani et al, 2017) | https://cryosparc.com; RRID:SCR_016501 |
| RELION 3.0.6 | (Zivanov et al, 2018) | http://www2.mrc-lmb.cam.ac.uk/relion/; RRID:SCR_016274 |
| UCSF Chimera 1.16 | (Pettersen et al, 2004) | https://www.rbvi.ucsf.edu/chimera/; RRID:SCR_004097 |
| UCSF ChimeraX 1.3 | (Pettersen et al, 2021) | https://www.rbvi.ucsf.edu/chimerax/; RRID:SCR_015872 |
| COOT 0.8.9 | (Emsley and Cowtan, 2004) | http://www2.mrc-lmb.cam.ac.uk/personal/pemsley/coot; RRID:SCR_014222 |
| AlphaFold | (Jumper et al, 2021) | https://alphafold.ebi.ac.uk/ |
| PHENIX 1.17.1 | (Afonine et al, 2018) | http://www.phenix-online.org/; RRID:SCR_014224 |
| Biorender | BioRender | https://app.biorender.com/ |
| PyMOL 2.5.0 | Schrodinger LLC | https://pymol.org/2/; RRID:SCR_000305 |
| Molprobity | (Davis et al, 2007) | http://molprobity.biochem.duke.edu |
| **Other** | | |
| FastPure Gel DNA Extraction Mini Kit | Vazyme | Cat# DC301-01 |
| Carbon coated copper grid | ZHONGJINGKEYI | CAT#BZ100205b |
| Quantifoil Cu R1.2/1.3 + 2 nm C, 300 mesh | Quantifoil | Cat# X-102-Cu300C2 |

## Generation of knock-in *Chlamydomonas* strains

The Cdc5L-Flag and Prp19-Flag KI strains were generated by CRISPR/Cas9-based gene editing following previous protocols with slight modifications (Ferenczi et al, 2017; Kelterborn et al, 2022; Shin et al, 2016). The sgRNA target sites were predicted by CRISPR-P (Lei et al, 2014) and the sgRNAs were commercially synthesized

(Genscript). The ribonucleoproteins (RNP) were individually assembled by pre-incubating the sgRNA with Cas9 protein from *Streptococcus pyogenes* at 37 °C for 20 min. Each double-stranded donor DNA (carrying a 3x Flag tag and 20 bp homology arms) and the paromomycin resistance expression cassette were amplified using PCR. The RNPs, donor DNA segments, and paromomycin resistance expression cassette were all delivered into the WT *Chlamydomonas* by electroporation. After initial screening on plates, the genotypes of the transformants were further verified by PCR and confirmed by DNA sequencing. The target sites for the sgRNA used for Cdc5L and Prp19 were 5′-ACTAGCGCAGAAGCGGGCGG-3′ and 5′-CCACA-TAACCGCTTAAGCGG-3′, respectively.

## Purification of the *Chlamydomonas* spliceosome

The Cdc5L-Flag and Prp19-Flag KI cell cultures were grown at 23 °C in Tris-acetate-phosphate (TAP) medium to an $OD_{750}$ of 0.5–0.6. The spliceosomes from the two genetically modified *Chlamydomonas* strains were purified following the same procedures at 4 °C (Fig. EV1A). The collected algal cell pellets were resuspended in a lysis buffer containing 20 mM HEPES-KOH pH 7.9, 150 mM NaCl, 1.5 mM $MgCl_2$, 4% glycerol (v/v), and protease inhibitors (1 mM PMSF, 2.6 μg/mL aprotinin, 1.4 μg/mL pepstatin, and 5 μg/mL leupeptin). The cell suspension was dropped into liquid nitrogen to form cell beads and pulverized into powder in a Retsch ZM200 nitrogen mill. The frozen cell powder was thawed at room temperature and resuspended in the same lysis buffer. The cell lysate was centrifuged at 13,000 rpm for 1 h, and the resulting supernatant was loaded into the Anti-FLAG M2 resin (Sigma). The protein-loaded resin was then rinsed in a wash buffer containing 20 mM HEPES-KOH pH 7.9, 150 mM NaCl, 1.5 mM $MgCl_2$, 4% glycerol (v/v), and 0.01% NP-40. The spliceosomal complexes were eluted by the lysis buffer, which was supplemented with 0.2 mg/mL FLAG peptide (DYKDDDDK). The eluent was concentrated and further applied to the linear 10–30% (v/v) glycerol gradient supplemented by 0–0.1% EM-grade glutaraldehyde. After centrifugation at $110,000 \times g$ for 15 h in a SW32 rotor (Beckman Coulter), the sample was manually fractionated into 19 fractions of about 2 mL each. After total RNA extraction, the RNAs from each fraction were extracted and analyzed on a denaturing 8% urea-polyacrylamide gel (Fig. EV1B,C). The fractions containing spliceosomal complexes were pooled, concentrated, and dialyzed against a D150 buffer (20 mM HEPES-KOH pH 7.9, 150 mM NaCl, 1.5 mM $MgCl_2$).

## Cryo-EM sample preparation and data collection

The isolated *Chlamydomonas* spliceosomes were checked by negative staining and subsequently used for cryo-sample preparation. Holey carbon grids coated with 2 nm carbon film (Quantifoil, Cu, 300-mesh, R1.2/1.3, 2 nm C) were glow-discharged in a plasma cleaner (HARRICK PLASMA Company). Each aliquot (4 μL) of the sample was applied to a glow-discharged grid and waited for 60 s before conducting a 3.5 s blotting step. The grid was then quickly plunged into liquid ethane cooled by liquid nitrogen using a Vitrobot Mark IV (Thermo Fisher) at 8 °C and 100% humidity.

For the Cdc5L-Flag dataset, the cryo-grids were imaged under a 300-kV Titan Krios electron microscope (G4, Thermo Fisher Scientific) equipped with a Falcon 4i detector and a GIF Quantum

energy filter (slit width 20 eV). Micrographs were captured at a normal magnification of 130,000× with a pixel size of 0.92 Å (Fig. EV1D). Each movie was exposed to a total dose of ~50 e-/Å² and automatically collected using EPU software (Thermo Fisher Scientific) with a preset defocus ranging from −1.0 to −1.5 μm. The micrographs were subsequently imported into cryoSPARC (Punjani et al, 2017) and applied for patch motion correction. The defocus value for each image was determined by Gctf (Zhang, 2016).

For the Prp19-Flag dataset, the cryo-grids were transferred to a 300-kV Titan Krios electron microscope (G3i) equipped with a Gatan K3 detector and a GIF Quantum energy filter (slit width 20 eV). Micrographs with a defocus range of −1.4 to −1.8 μm were collected at a normal magnification of 81,000× in the super-resolution mode with a physical pixel size of 1.077 Å using EPU software (Fig. EV1E). Each stack was automatically exposed for 2.56 s, with 32 frames and a total dose of ~50 e-/Å². The movies were aligned and summed using MotionCor2 (Zheng et al, 2017), with dose weighting performed concurrently. The defocus value for each micrograph was determined by Gctf (Zhang, 2016).

## Cryo-EM data processing

Preliminary cryo-EM data analysis was performed using cryoSPARC on a small dataset of the collected micrographs. Two-dimensional (2D) classification results for both the Cdc5L-Flag and Prp19-Flag datasets showed distinct features and initial 3D volumes were generated using select good particles (Fig. EV1F,G). The simplified procedures for cryo-image data processing of the full Cdc5L-Flag dataset are shown in Fig. EV2. In total, 15,813 good micrographs were manually selected from 17,343 collected images and about 1.7 million particles were generated by Gautomatch (developed by Kai Zhang, https://www2.mrc-lmb.cam.ac.uk/download/gautomatch-053/). All the particles at bin-4 level (pixel size: 3.68 Å) were applied to global and local 3D classifications, resulting in about 700 thousand good particles, which were selected and re-extracted at bin-2 level (pixel size: 1.84 Å) and reconstructed at 3.7 Å. These good particles were further re-centered and re-extracted at an unbinned level (pixel size: 0.92 Å), and applied for Skip_alignment 3D classifications in RELION (Zivanov et al, 2018). The remaining 518,369 selected particles were imported into cryoSPARC and Non-Uniform refinement (Punjani et al, 2020) was performed, which generated a reconstruction at a resolution of 2.6 Å. To improve the local EM map quality in the peripheral regions, soft masks were applied to the U5 snRNP, the N-terminus of Prp17, and the Prp19 subcomplex, generating three local EM maps at resolutions of 3.6 Å, 3.0 Å, and 3.6 Å, respectively (Fig. EV2).

For the Prp19-Flag dataset, a total of 7328 micrographs were collected, and 7259 good images were manually selected for further processing (Fig. EV3). Two different batches of particles were automatically picked using Gautomatch, with the particles in the second batch auto-picked by excluding the bin-1 level particles from the first batch. The processing of both batches of particles was almost the same as for the Cdc5L-Flag dataset. Finally, 521,250 good particles were collected after the removal of duplicates. These particles were reconstructed in cryoSPARC using Non-Uniform refinement and generated a map at a resolution of 2.6 Å, which is almost identical to that from the Cdc5L-Flag dataset. Local soft masks were further applied to the N-terminus of Prp17 and the Prp19 subcomplex, improving local resolutions to 3.5 Å and 3.4 Å, respectively (Appendix Table S1).

Reported resolutions were calculated based on an FSC value of 0.143 (Chen et al, 2013) (Fig. EV4A,B). Local resolution variations were estimated using cryoSPARC (Fig. EV4C,D). The angular distributions of the particles used for the final reconstructions are well-distributed (Fig. EV4E).

## Model building and refinement

The atomic model for the *Chlamydomonas* C* complex was de novo built based on our high-resolution EM maps using COOT (Emsley and Cowtan, 2004) (Appendix Table S2). The available model of the human C* complex (Zhan et al, 2022) (PDB: 7W5B) was fitted into the *Chlamydomonas* EM maps using Chimera (Pettersen et al, 2004), and the individual protein components and RNA elements were initially identified. Modeling of the identified protein components was facilitated by the predicted structures of Alpha-Fold (Jumper et al, 2021), and these structures were fitted into the EM maps and manually adjusted using COOT (Emsley and Cowtan, 2004). The RNA elements were manually checked and mutated based on those in the human C* complex (Zhan et al, 2022) (PDB: 7W5B).

The final model of the *Chlamydomonas* C* complex was refined according to our high-quality EM maps using the phenix.real_space_refine program in PHENIX with secondary structure restraints (Afonine et al, 2018). Overfitting of the model was monitored by refining the model in one of the two independent maps using the gold-standard refinement approach and testing the refined model against the other (Amunts et al, 2014) (Fig. EV4F). The structure was further validated by examination of their Molprobity scores and statistics from Ramachandran plots. The Molprobity scores were calculated as described (Davis et al, 2007). The EM maps clearly display the features of the nucleobases of RNA elements and the side chains of the protein components (Fig. EV5; Appendix Figs. S1–3), allowing detailed structural examination of the interplay between proteins and RNAs.

# Data availability

The atomic coordinates for *Chlamydomonas* C* complex have been deposited in the Protein Data Bank (PDB, https://www.rcsb.org/) under accession code 8XI2. The overall EM maps of *Chlamydomonas* C* complex have been deposited in the Electron Microscopy Data Bank (EMDB, https://www.ebi.ac.uk/emdb/) with accession codes EMD-38362 and EMD-38366 for the Cdc5L-Flag and Prp19-Flag datasets, respectively. The local refined EM maps have been deposited in the EMDB with accession codes EMD-38363, EMD-38364, EMD-38365, EMD-38367, and EMD-38368.

The source data of this paper are collected in the following database record: biostudies:S-SCDT-10_1038-S44318-024-00274-3.

# Peer review information

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

## Acknowledgements

We thank prof. Yigong Shi for the great support and suggestions. We thank the Cryo-EM Facility and the High-Performance Computing Center at Westlake University for providing cryo-EM and computation support. We thank the Mass Spectrometry & Metabolomics Core Facility of Westlake University for their support with the mass spectrometry experiments and analyses. This work was supported by funds from the National Natural Science Foundation of China (31930059).

## Author contributions

**Yichen Lu**: Data curation; Software; Formal analysis; Validation; Investigation; Visualization; Methodology; Writing—original draft. **Ke Liang**: Resources; Data curation; Software; Investigation; Visualization; Methodology; Writing—original draft. **Xiechao Zhan**: Conceptualization; Resources; Data curation; Software; Formal analysis; Supervision; Validation; Investigation; Visualization; Methodology; Writing—original draft; Writing—review and editing.

Source data underlying figure panels in this paper may have individual authorship assigned. Where available, figure panel/source data authorship is listed in the following database record: biostudies:S-SCDT-10_1038-S44318-024-00274-3.

## Disclosure and competing interests statement

The authors declare no competing interests.

# Expanded View Figures

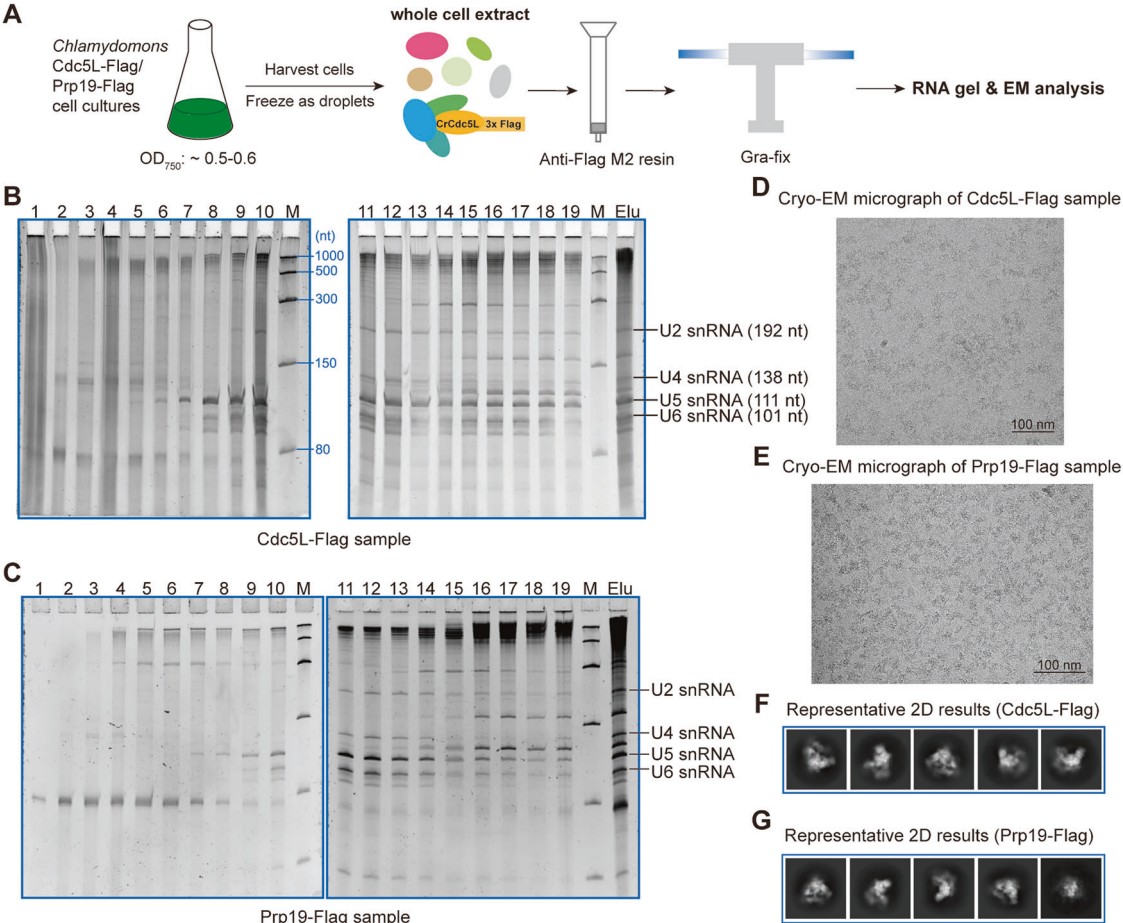

**Figure EV1.  Purification and analysis of *Chlamydomonas* spliceosomal complexes.**

(**A**) A simplified schematic diagram for the isolation of *Chlamydomonas* spliceosomal complexes. (**B**, **C**) The putative spliceosomal complexes were analyzed after glycerol gradient centrifugation with chemical crosslinking using urea-PAGE gels from the *Chlamydomonas* Cdc5L-Flag (**B**) and Prp19-Flag (**C**) strains. (**D**, **E**) Representative cryo-EM micrographs of the *Chlamydomonas* spliceosomal complexes from the Cdc5L-Flag (**D**) and Prp19-Flag (**E**) strains. Scale bar, 100 nm. (**F**, **G**) Representative 2D average results of the *Chlamydomonas* spliceosomal complexes from the Cdc5L-Flag (**F**) and Prp19-Flag (**G**) datasets.

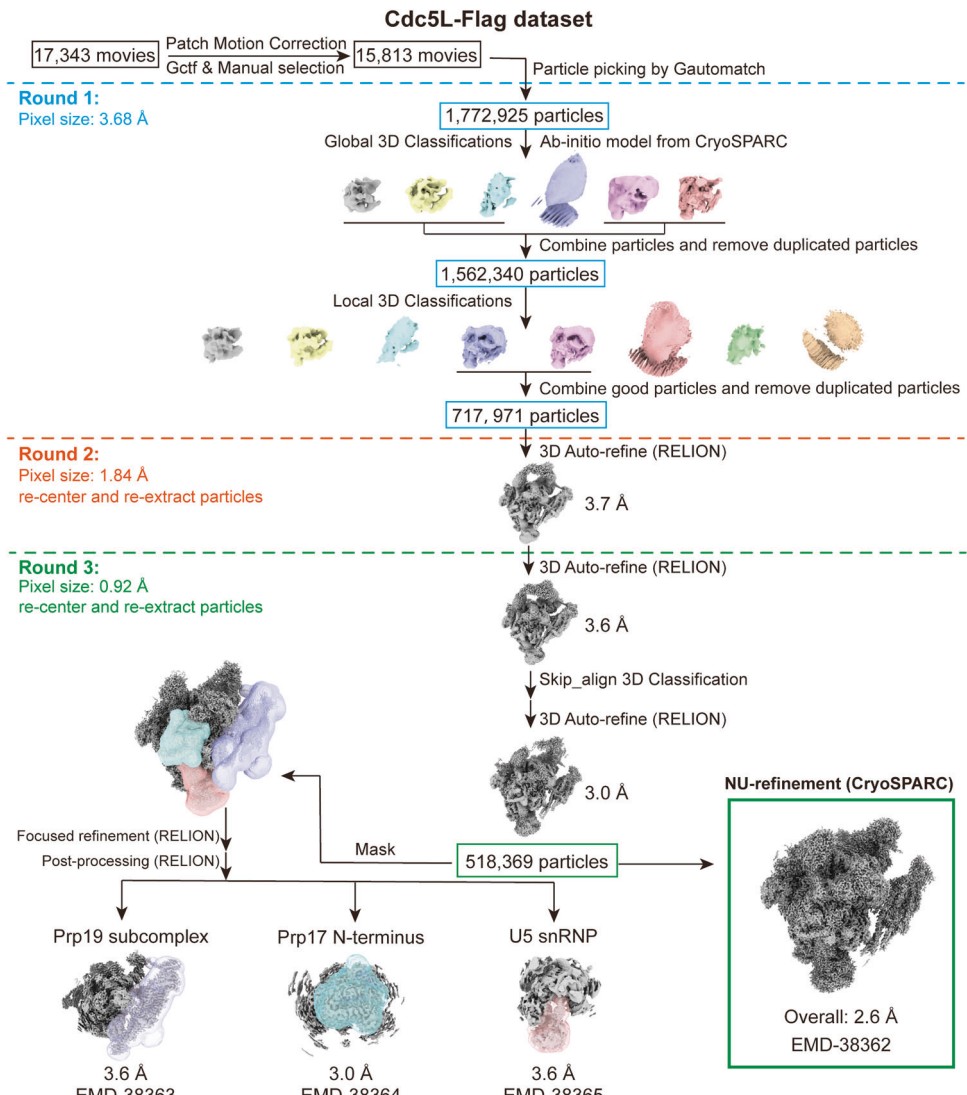

**Figure EV2.   A flow chart of cryo-EM data processing for the *Chlamydomonas* C* complex from the Cdc5L-Flag dataset.**

All processing steps were carried out in RELION 3.0 and cryoSPARC. Please refer to Methods for details.

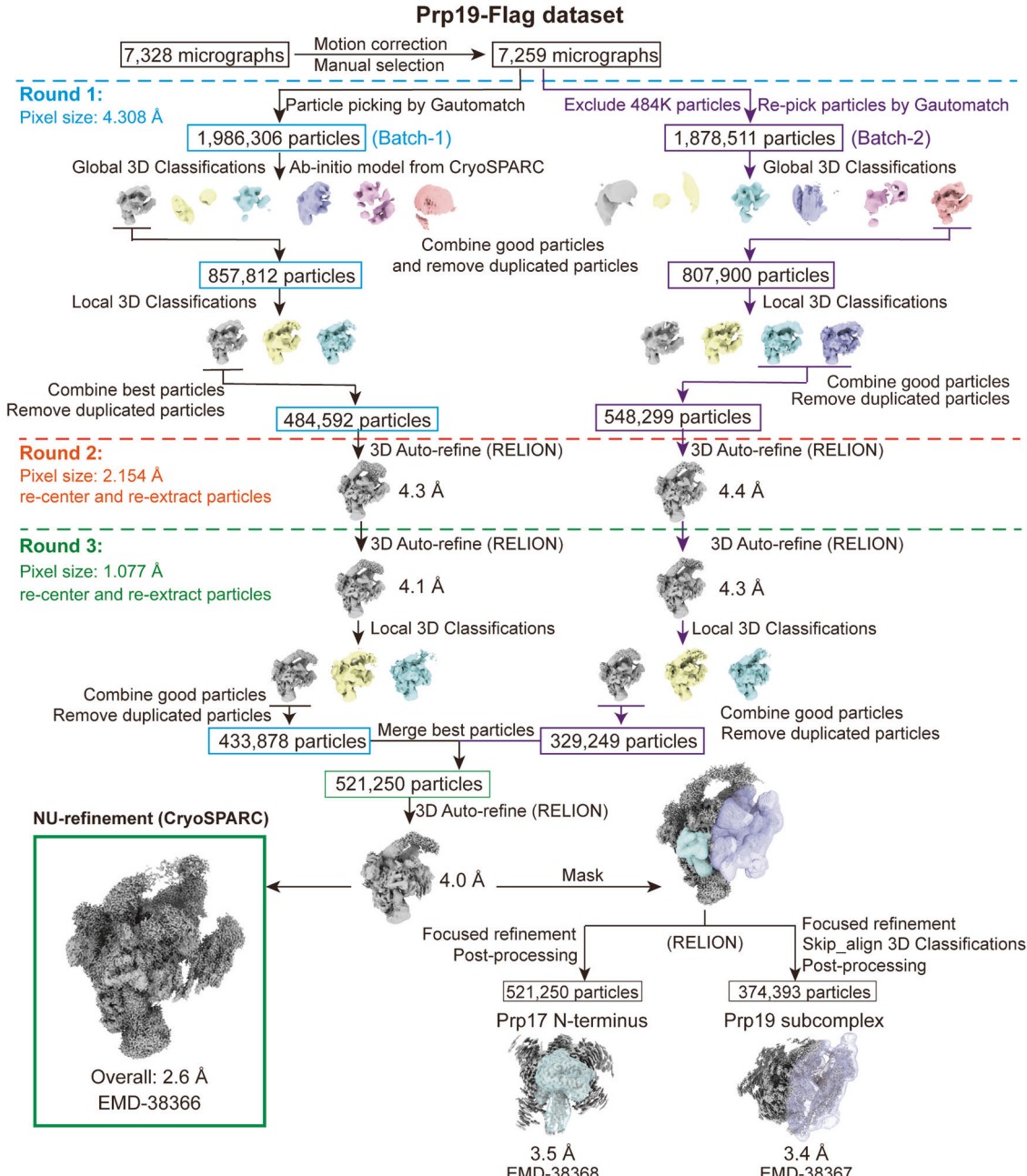

**Figure EV3. A flow chart of cryo-EM data processing for the *Chlamydomonas* C* complex from the Prp19-Flag dataset.**

All processing steps were carried out in RELION 3.0 and cryoSPARC. Please refer to Methods for details.

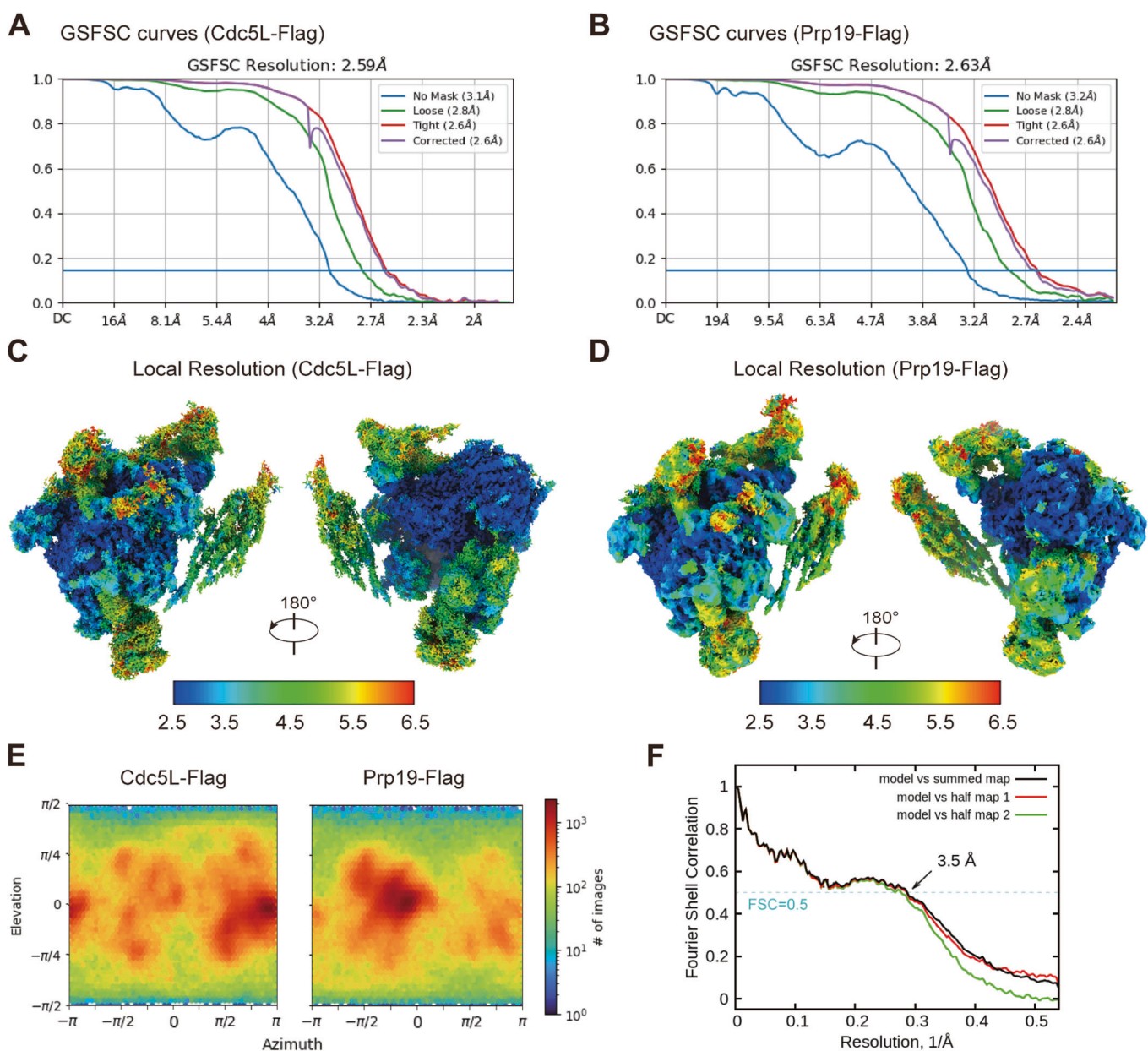

**Figure EV4.  Cryo-EM reconstruction of the *Chlamydomonas* C* complex.**

(A, B) Shown here are the FSC curves for the final refinement in cryoSPARC. The average resolutions have been both achieved at 2.6 Å for the *Chlamydomonas* C* complex using the Cdc5L-Flag (A) and Prp19-Flag (B) datasets. (C, D) Shown here are the local resolutions color-coded for different regions from the reconstructions using the Cdc5L-Flag (C) and Prp19-Flag (D) datasets. (E) Shown here are the angular distributions of the particles used for the final reconstructions in cryoSPARC for the Cdc5L-Flag (left panel) and Prp19-Flag (right panel) datasets. (F) The FSC curves for the cross-validation between the model and the cryo-EM maps of the *Chlamydomonas* C* complex. Shown here are the FSC curves between the final refined atomic model and the reconstruction from all particles (black), between the model refined in the reconstruction from only half of the particles and the reconstruction from that same half (red), and between that same model and the reconstruction from the other half of the particles (green).

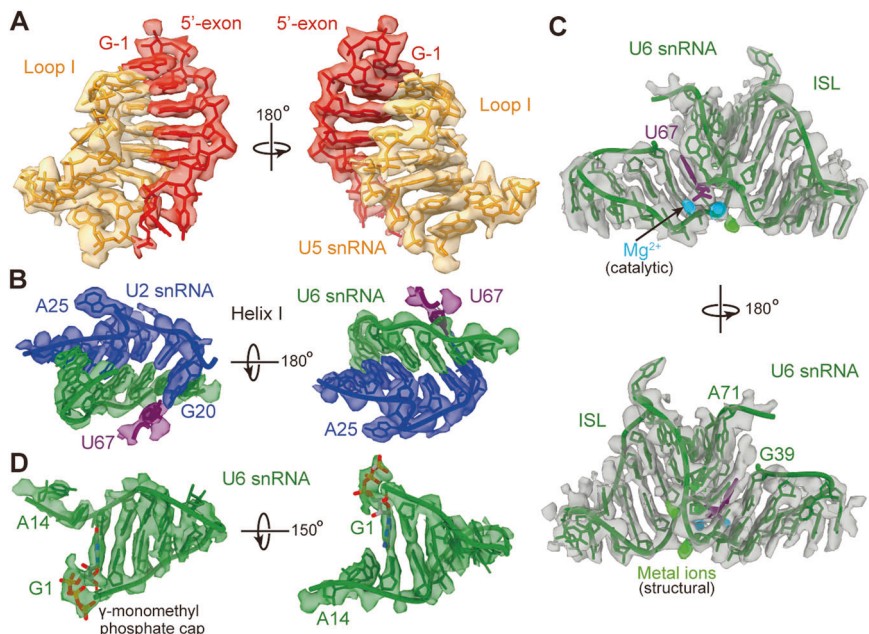

**Figure EV5. Representative EM maps of the RNA elements.**

(A) The EM maps of the 5'-exon and loop I of U5 snRNA. Two related views are shown. (B) The EM maps of the Helix I. Two related views are shown. (C) The EM maps of the ISL of U6 snRNA. Two catalytic metal ions are coordinated by the key nucleotide U67. Two related views are shown. (D) The EM maps of the 5'-SL of U6 snRNA. Two related views are shown.

