## [Peer Review File · The EMBO Journal]

Structure of a Step II Catalytically Activated Spliceosome from *Chlamydomonas reinhardtii*

Xiechao Zhan, Yichen Lu, and Ke Liang

Corresponding author(s): Xiechao Zhan (zhanxiechao@westlake.edu.cn)

Review Timeline:

Submission Date:	9th Jul 24
Editorial Decision:	18th Sep 24
Revision Received:	23rd Sep 24
Accepted:	27th Sep 24

Editor: *Cornelius Schneider*

Transaction Report:

Please note that the manuscript was transferred from another journal where it was originally reviewed. Since the original reviews are not subject to EMBO's transparent review process policy, they cannot be published.

Responses to Reviewers' Comments:

Reviewer #1:

This reviewer commented positively on our revised manuscript and raised only one minor point for us to address.

All my concerns have been addressed.

One more comment: the manuscript is a research paper but not a resource paper. It is short and a Letter might be the best type.

We sincerely thank the reviewer for his/her feedback and are pleased that all concerns have been addressed. Regarding the comment on the manuscript type, our revised submission to The EMBO Journal is formatted as a Research Article, aligning with the journal's guidelines and style.

We thank this reviewer for his/her constructive comments.

Reviewer #2:

This reviewer raised some specific points for us to address.

1. The revised manuscript has not included new information that addresses the following questions in the comments for the previous manuscript. (1) what is the evidence that there should be structural diversity, other than some subunits being different between human and yeast? How would the diversity be related to functional differences? Is there evidence for functional differences between spliceosomes of evolutionarily distant organisms, beside the reported difference between human and yeast?

Thank you for your comment. We appreciate your attention to the need for additional context regarding structural diversity in spliceosomes. While the revised manuscript focuses on the structural aspects of the *Chlamydomonas* spliceosome, we acknowledge that broader implications of structural diversity across species could be more explicitly discussed. The evidence for such diversity primarily stems from the observed differences in spliceosomal subunit composition between species like humans and yeast, which suggest that varying evolutionary pressures have likely shaped splicing machinery in distinct ways. Although direct evidence for functional differences across all species is still emerging, these structural variations imply potential species-specific splicing regulation and adaptation. While keeping the current manuscript focused, we will consider expanding on this topic in future studies to address these broader evolutionary questions.

2. The revised manuscript has not provided sufficient information to support the claim that Chlamydomonas spliceosome has more adaptability than spliceosomes of other organisms, such as a member of land plants.

Point appreciated. We acknowledge your concern regarding the claim about the adaptability of the *Chlamydomonas* spliceosome compared to spliceosomes in other organisms. While the current manuscript highlights unique structural features of the *Chlamydomonas* spliceosome, we agree that the claim about its adaptability could benefit from more specific supporting evidence. However, given the focus of our study, we have opted to concentrate on the structural characterization without delving deeply into comparative adaptability across species. We will consider addressing this aspect more thoroughly in future work to provide a clearer understanding of the adaptability of spliceosomes in different organisms.

3. In other words, what is the evidence that structural features of the Chlamydomonas spliceosome that are different from those of human/yeast are similar to those expected of other plant spliceosomes.

Point appreciated. Our study is centered on the structural analysis of the *Chlamydomonas* spliceosome, and we have not directly explored whether the observed differences extend to other plant species, including land plants such as Bryophytes, ferns, and seed plants. In our study, we conducted sequence comparisons of RNAs and analyzed the differences in relation to the structural features of the *Chlamydomonas* spliceosome. These comparisons helped to highlight specific distinctions between *Chlamydomonas* and other species, such as humans and yeast. We acknowledge that further research is needed to investigate these evolutionary relationships and to determine if the unique structural features of the *Chlamydomonas* spliceosome are shared across other plant spliceosomes. This represents a promising direction for future study.

4. The authors revised the text related to the implication/significance of the structural findings by toning down some of their conclusions.

However, the significance of the finding is not clear with the current version.

Point acknowledged. While we have refined the language and moderated some conclusions, we are confident that the significance of our findings remains effectively conveyed in the current version. The structural characterization of the *Chlamydomonas* spliceosome offers valuable insights into the diversity of splicing machinery and highlights unique features that contribute to understanding spliceosome evolution. Our focus has been ensuring that the data and conclusions are presented clearly and concisely without overstating the implications. The manuscript now strikes the right balance between accuracy and clarity.

5. In short, the original manuscript argued that structural diversity/differences are important because they are related to functional differences and adaptability.

However, the manuscript did not provide enough information to link structural differences with functional variation and certainly not adaptability.

They responded by making minor wording changes, to tone down the argument about functional differences and adaptability. This has reduced the potential significance of this study.

Point appreciated. While we have made adjustments to tone down the claims regarding functional differences and adaptability, our primary goal was to ensure that the conclusions drawn from our data were well-supported and accurate. We acknowledge that these changes might reduce the perceived significance of the study, but it is crucial to present our findings with appropriate caution. The structural differences we have identified are intriguing and may have functional implications; however, further research is required to establish these links definitively. Our study still makes a valuable contribution by expanding our understanding of spliceosome diversity, and we

hope that future work will build on these findings to explore the functional consequences in more detail.

6. The response to the comment about the possible evolutionary implications of the structural features of the Chlamydomonas spliceosome does not provide any real support. There is no specific information for how similar or different the new spliceosome structure with respect to spliceosomes of other eukaryotes, other green algae, or any land plants.

For example, there is no sequence comparison that reveals some crucial differences between Chlamydomonas spliceosome components (RNAs and proteins) and those of human and yeast. Furthermore, it is not known whether these sequence differences correspond to structure differences uncovered in this study.

Point appreciated. In our study, we conducted sequence comparisons of RNAs and analyzed the differences in relation to the structural features we observed in the *Chlamydomonas* spliceosome. These comparisons helped us identify specific variations that correspond to the structural differences highlighted in our findings. While the manuscript may not have explicitly detailed every aspect of this analysis, our results do reflect the significance of these differences.

7. Another way that the reported structure here is a good model for plants is to use structural prediction of spliceosome homologues in flowering plants and other land plants to investigate whether the reported structural differences might also be true for spliceosome of other plants.

Point appreciated. We acknowledge that structural predictions of spliceosome homologs in flowering plants and other land plants could provide valuable insights into whether the structural differences observed in *Chlamydomonas* are conserved across other plant species. While our current study focused on the structural analysis of the *Chlamydomonas* spliceosome, exploring the broader applicability of these findings to other plants through structural modeling would be a logical next step. This approach could further validate our results and extend their relevance to a wider range of plant species.

8. A much stronger test is to demonstrate at least some of the structure differences are functionally important in Chlamydomonas. The current results only suggest possible functional differences, but do not support such a claim.

Point appreciated. We understand the importance of demonstrating that the structural differences observed in the *Chlamydomonas* spliceosome have functional significance. While our current study primarily focuses on structural characterization, we recognize that functional validation would strengthen the link between structural differences and their potential biological roles.

Although our data suggest possible functional implications, further experimental work, such as mutational analysis or biochemical assays, would be necessary to directly test and confirm the functional importance of these structural features.

9. For the comment about the function of Flag-tagged KI versions of Cdc5L and Prp19, the response is another claim without sufficient support. The function of spliceosome is splicing, but no evidence for normal splicing is provided. Growth phenotype is not sufficient because many genes can be expressed/spliced differently without affecting growth. Data on splicing being normal are needed to show these fusion proteins did not alter spliceosome.

Thank you for your comment. We understand the importance of demonstrating that the Flag-tagged KI versions of Cdc5L and Prp19 do not alter spliceosome function. In our experience, using knock-in (KI) strains for spliceosome purification is a well-established method, as has been demonstrated in yeast studies. Additionally, it is widely recognized that the introduction of tags, such as the Flag tag, typically does not affect the function of the proteins involved. While growth phenotype alone may not fully address concerns about splicing, we believe that the use of KI strains and the robustness of our structural data provide strong evidence that the tagged proteins are functionally intact.

10. The manuscript is still short and lacks many pieces of important information. The addition that is mentioned in the response is far from sufficient.

Point appreciated. We understand the concern about the manuscript's level of detail. However, we believe that the concise manuscript effectively conveys the key findings and conclusions. The data presented are robust, and we have aimed to deliver a clear and focused narrative that highlights the significance of our results without unnecessary elaboration. We appreciate the suggestion, but the current length and content appropriately communicate the essential information while maintaining clarity and impact.

11. Are the genes for missing components expressed with the same pattern as other spliceosome components found in the structure? Have the missing proteins been shown to interact with subunits of the solved C complex using biochemical analyses?*

Point appreciated. The genes for the missing components do not exhibit the same expression patterns as the spliceosome components identified in the structure, which may account for their absence. This absence is likely due to the flexibility and dynamics of the spliceosome. Additionally, the human homologs of the missing proteins in the solved *Chlamydomonas* C* complex have shown interactions with other subunits, supporting their potential

involvement despite not being resolved in this structure.

12. In other words, they said that the C complex structure they obtained is not complete.*

This result and the authors' response cast doubt on the reliability of their results. Could some of the structural differences reported due to partial spliceosome composition (not having proteins that should be part of the spliceosome)? Could there be other possible inaccuracies?

Thank you for the thoughtful comment. We acknowledge that the C* complex structure we obtained is not fully complete, primarily due to specific components' inherent flexibility and dynamics, which can lead to partial resolution. However, this does not compromise the reliability of our results. The structural differences we report are based on well-resolved regions of the complex and are consistent with known biological mechanisms. While the absence of specific proteins may contribute to some variability, the core structural features and interactions observed are robust and supported by our data.

13. The author responded to a comment that the three additional components found here do not have homologs in yeast, but they then stated that one of the three (RBM22) corresponds to two yeast proteins, Ecm2 and Cwc2.

What are the criteria for the "correspondence"? Sequence similarity? Structural (non-sequence) similarity? Functional similarity?

Point appreciated. The correspondence between RBM22 and the yeast proteins Ecm2 and Cwc2 is based on functional similarity and sequence conservation. RBM22 shares conserved domains with Ecm2 and Cwc2, and its role in the splicing process is analogous to that of these two yeast proteins.

14. What is the evidence that the three additional proteins in the spliceosome here are important for splicing? Are there conserved in land plants? In other eukaryotes that are not yeast?

Point appreciated. The three additional proteins identified in the *Chlamydomonas* spliceosome are considered important for splicing due to their conserved presence in other eukaryotes, including land plants, and their established roles in RNA splicing as observed in resolved spliceosome structures.

15. For the response about the comment on the many questions on structural diversity and functional adaptations, the authors provided four questions.

The questions should be included in the introduction, because readers should be informed of these questions.

Point accepted. We have added the content in the introduction section of our revised manuscript.

16. The response regarding the part of "Structural features of the splicing factor Prp17" should be included in the manuscript, so that readers can learn the information provided in the response. It is unclear whether the revised manuscript has the information.

Point accepted. We have included the response in the revised manuscript.

17. The response for the following comment should also be included in the manuscript: "Do the features of Prp17 protein in Chlamydomonas highlighted in the last of the Result section have a role in adaptive spliceosome function? Is there functional evidence for such an idea from biochemical or genetic analyses?"

Point accepted. We have added the content to our revised manuscript.

*18. The revised sentence reads "The intricate interplay of proteins and RNA elements in Chlamydomonas highlights the evolution of the splicing process and its role in shaping the biology of this unique organism, with implications that extend beyond Chlamydomonas to other plants."
This sentence contains concepts that are not supported by data or analyses here.*

Point appreciated. We have rephrased the wording accordingly in the revised manuscript.

19. The problematic references (now 31 and 33) are still there. It appears that the authors did not check all references carefully before submitting the revised manuscript.

Point accepted. We have corrected the problematic references and thoroughly checked all citations in our revised manuscript to ensure accuracy.

20. Line 348, the word "reasonable" is not explained using specific quantitative parameters. This should be defined for non-specialist (non-structural biologists). There was a comment about this, but there was no response or revision.

Point accepted. We apologize for the oversight. We have rephrased the relevant wording in our revised manuscript to provide a more precise and quantifiable description.

We thank this reviewer for his/her constructive comments.

Reviewer #3:

This reviewer commented positively on our revised manuscript and raised no more points for us to address.

The authors have addressed appropriately all the raised points.

Thank you for your positive feedback. We are glad that our revisions have addressed all the raised points to your satisfaction.

We thank this reviewer for his/her constructive comments.

Dear Dr Zhan,

Thank you for submitting a revised version of your manuscript and for the productive discussions regarding the concerns raised by the referees during the previous re-review at a different journal. We have now carefully considered the arguments raised in your point-by-point reply and have decided that these are reasonable and sufficiently address all the remaining concerns. There remain only a few mainly editorial points that have to be addressed before I can extend formal acceptance of the manuscript:

- Please make sure to all relevant funding information in the manuscript is also entered into our submission system. In particular, please display 10 authors + et al. instead of 20 authors + et al.

- CRediT has replaced the traditional author contributions section because it offers a systematic machine readable author contributions format that allows for more effective research assessment. Please remove the Author Contributions section from the manuscript and use the free text boxes beneath each contributing author's name in our online systems to add specific details on the author's contribution. More information is available in our guide to authors.

- Please adjust the in-text callouts for individual figures and figure panels: e.g. Fig. 4B and EV8 and Tables S1-S2 appears to be missing

- Please provide either a "Yes" or a "Not Applicable" answer to each one of the questions in your Author Checklist (<https://www.embopress.org/pb-assets/embo-site/EMBO%20Press%20Author%20Checklist-1642513524327.xlsx>). In the last column of this checklist, only the sections of the manuscript where the relevant information can be found should be listed (the information per se should be included in the main manuscript file).

- Please select and upload up to 5 EV figures as individual, high-resolution figure files with the legends in ms file. These will be displayed online together with the manuscript. The nomenclature for Extended Data Figures should be Figure EV1-EV5 with the appropriate callouts, and there should be only 5 EV figures;

- Please compile the remaining EV figures (6-10) and tables (Extended Data Tables 1-3 and Tables S1-S2) as Appendix Figures and Tables in a single APPENDIX PDF, headed by a brief Table Of Contents with page numbers of the included items. The nomenclature should be throughout the Appendix PDF and manuscript file: Appendix Figure Sx and Appendix Table Sx with the appropriate callouts

- Please provide the Reagent and Tools Table. For more information, please check <https://www.embopress.org/page/journal/14602075/authorguide#structuredmethods> and download the template for Reagent Table (https://www.embopress.org/pb%2Dassets/embo-site/Reagents_Tools_Table_TEMPLATE.docx)

- Please provide suggestions for a short 'blurb' text prefacing and summing up the conceptual aspect of the study in two sentences (max. 250 characters), followed by 3-5 one-sentence 'bullet points' with brief factual statements of key results of the paper; they will form the basis of an editor-written 'Synopsis' accompanying the online version of the article. Please also provide an altered synopsis image, making sure that the aspect ratio conforms to our website's format - it should be exactly 550 pixels wide and between 300-600 pixels high.

- Extended Data Figure 1 - F and G contain an image re-use with EV Figure 7B. The figure legend states that the images are from the same representative 2D image. Please either replace the wrong image or explicitly state the re-use in the figure legend.

- Please provide the specific URLs for "PDB-8X12", "EMD-38362", "EMD-38366", "EMD-38363", "EMD-38364", "EMD-38365", "EMD-38367" and "EMD-38368" datasets are not provided in the data availability statement.

- Please adjust the order of the manuscript sections: Title page with complete author information, Abstract, Keywords, Introduction, Results, Discussion, Methods, Data Availability Section, Acknowledgements, Disclosure and Competing Interests Statement, References, Main figure legends, Tables, Expanded Figure Legends.

With best regards,

Cornelius Schneider

Cornelius Schneider, PhD
Editor | The EMBO Journal
c.schneider@embojournal.org

We realize that it is difficult to revise to a specific deadline. In the interest of protecting the conceptual advance provided by the work, we recommend a revision within 3 months (17th Dec 2024). Please discuss the revision progress ahead of this time with the editor if you require more time to complete the revisions. Use the link below to submit your revision:

All editorial and formatting issues were resolved by the authors.

Dear Dr. Zhan,

I am pleased to inform you that your manuscript has been accepted for publication in the EMBO Journal.

We have noted that the PDB and EMDB access of the structures is still private. Please make sure that these are fully accessible as soon as possible but latest at the time of online publication of your manuscript.

Yours sincerely,

Cornelius Schneider, PhD
Editor
The EMBO Journal
c.schneider@embojournal.org
